# Research on a New Intelligent and Rapid Screening Method for Depression Risk in Young People Based on Eye Tracking Technology

**DOI:** 10.3390/brainsci13101415

**Published:** 2023-10-05

**Authors:** Zhanbo Tao, Ningxia Sun, Zhen Yuan, Zeyuan Chen, Jiakang Liu, Chen Wang, Shuwu Li, Xiaowen Ma, Bin Ji, Kai Li

**Affiliations:** 1Police Sports Department, Zhejiang Police College, Hangzhou 310053, China; Taozhanbo@zjjcxy.cn; 2Joint Laboratory of Police Health Smart Surveillance, Zhejiang Police College, Hangzhou 310053, China; czy2451923029@163.com; 3Department of Reproductive Medicine, Second Affiliated Hospital of Naval Medical University, Shanghai 200003, China; suesunchzh@126.com; 4Centre for Cognitive and Brain Sciences, University of Macau, Macau SAR 999078, China; zhenyuan@um.edu.mo; 5Zhejiang-Japan Digital Diagnosis and Treatment and Equipment of Integrated Traditional Chinese Medicine and Western Medicine for Major Brain Diseases Joint Laboratory, Zhejiang Chinese Medical University, Hangzhou 310053, China; ljk06262022@163.com (J.L.); chenw773560532@163.com (C.W.); bookfiveli@outlook.com (S.L.); ma17781141732@163.com (X.M.); 6Department of Radiopharmacy and Molecular Imaging, School of Pharmacy, Fudan University, Shanghai 200032, China

**Keywords:** depression, young adults, affective computing, eye movement tracking technique, screening

## Abstract

Depression is a prevalent mental disorder, with young people being particularly vulnerable to it. Therefore, we propose a new intelligent and rapid screening method for depression risk in young people based on eye tracking technology. We hypothesized that the “emotional perception of eye movement” could characterize defects in emotional perception, recognition, processing, and regulation in young people at high risk for depression. Based on this hypothesis, we designed the “eye movement emotional perception evaluation paradigm” and extracted digital biomarkers that could objectively and accurately evaluate “facial feature perception” and “facial emotional perception” characteristics of young people at high risk of depression. Using stepwise regression analysis, we identified seven digital biomarkers that could characterize emotional perception, recognition, processing, and regulation deficiencies in young people at high risk for depression. The combined effectiveness of an early warning can reach 0.974. Our proposed technique for rapid screening has significant advantages, including high speed, high early warning efficiency, low cost, and high intelligence. This new method provides a new approach to help effectively screen high-risk individuals for depression.

## 1. Introduction

Depression is one of the most common mental disorders. According to the latest statistics from the World Health Organization, there are more than 350 million people with depression in the world, and the average global incidence is about 4.4%. It is estimated that depression will become the world’s largest burden of disease by 2030 [1,2]. The incidence of depression is age-specific, with young people being the main high-risk group for depression [3]. This is because they are in a critical transition period and need to cope with a variety of pressures, such as academic stress, career choice, financial burdens, and interpersonal and family stress [4]. Depression seriously affects their physical and mental health, personal development, and social life, leading to a serious decline in their quality of life. Patients with severe depression may even have harmful behaviors such as suicidal tendencies and self-injury. Worldwide, suicide caused by depression is the fourth leading cause of death among young people. Therefore, early screening for depression in young people is of great significance to reduce the incidence of depression and improve mental health.

At present, the gold standard for the diagnosis of depression remains a structured clinical interview, mainly through the observation, listening, and questioning of patients by psychiatrists and supplemented by rating scales. Self-report measures such as the Patient Health Questionnaire-9 (PHQ-9) and the Self-Rating Depression Scale (SDS) continue to play an important role in screening and measuring progress [5,6]. However, the above assessment methods have some shortcomings, such as high subjectivity and low accuracy. Coupled with the particularity of depression and the sensitive psychology of young people, they may selectively hide their illness because of psychological resistance. All of these factors make it difficult for many young people at high risk of depression to identify themselves and lose the opportunity for early intervention. Therefore, it is necessary to find an objective, rapid, intelligent, and accurate screening method for depression.

According to cognitive models associated with depression, individuals with depression tend to show the prioritized processing of materials that are consistent with their current mood in various cognitive tasks. This means that they typically display negative biases in various aspects of information processing, including interpretation, memory, and attention [7]. Attention bias to emotional stimulation is not only the core feature of depression in patients but is also an important factor affecting the onset, maintenance, and recurrence of depression [8,9,10]. The emotional attention bias of patients with depression primarily manifests as an increase in attention to negative emotional stimuli while also decreasing attention to positive emotional stimuli [11,12,13]. In addition, many studies have found that patients with depression exhibit attention bias in their facial features, specifically avoiding prominent facial features such as the eyes, nose, and mouth [14,15,16]. Based on the above characteristics of attention bias in patients with depression, the dynamic evaluation of facial emotional attention bias and facial feature attention bias in patients with depression may be a new way to promote the early warning and accurate screening of depression.

However, it is difficult to accurately and dynamically assess the attention bias of patients with depression through traditional structured clinical interviews and self-report measures. To provide an early warning for patients with depression more objectively, intelligently, and accurately, the application of “digital biomarkers” under specific tasks captured by digital equipment in the field of depression screening is attracting the attention of more and more scholars [17,18,19]. Digital biomarkers are defined as objective, quantifiable physiological and behavioral data that are collected and measured through digital devices [20]. Eye tracking technology has been explored as a digital screening technology that can accurately evaluate attention bias and has unique advantages in the field of affective computing. The relevant digital biomarkers extracted by the eye movement tracking technique also show good specificity in the early identification and screening of depression [21].

As a method that can directly measure attention distribution, eye tracking technology can provide non-invasive real-time data, visually analyze the fixation point and trajectory of eye movements, and effectively evaluate attention allocation and selective attention. Therefore, it can more objectively and effectively detect the characteristics of attention bias in depressed patients, thus realizing the early identification and screening of depression. Takahashi et al. [22] made a discriminant analysis using the scan path length and peak scan speed in the smooth tracking test and the free viewing test to distinguish depressed patients from healthy controls with 72.1% accuracy. To further improve the classification accuracy, Stolicyn et al. [23] predicted individual subjects’ depressive symptoms by combining facial and eye movements, and the detection accuracy reached 79%. Zhang et al. [24] used discriminant eye movement features such as scanning amplitude, scanning path length, and scanning speed extracted from multiple eye movement tasks to effectively distinguish depressed patients from healthy controls with 86% accuracy. The above research applies eye movement tracking technology to affective computing, which effectively promotes the current depression screening model to be more objective, accurate, and fine-grained. The new paradigm of dynamic perception evaluation that combines affective computing and eye movement tracking technology provides new ideas and methods for depression screening.

Based on previous views, we propose the following hypothesis: “Emotional perception of eye movement” may characterize defects in emotional perception, recognition, processing, and regulation in young people at high risk of depression. The evaluation paradigm based on “emotional perception of eye movement” is expected to become a new way of objective and rapid screening in young people at high risk of depression. Based on these above assumptions, we propose a novel, intelligent, and rapid screening method for depression risk in young people based on eye tracking technology. The aim of the study is to use this method to dynamically and intelligently capture the whole process of eye movement performed in the human–computer interaction (HCI) eye movement perception task in young people through the “eye movement emotional perception evaluation paradigm”. And finally, this study aims to achieve the accurate and rapid intelligent screening of young people at high risk for depression in an environment without doctors.

## 2. Materials and Methods

### 2.1. Participants

We recruited 100 first-year undergraduates aged 18–22 (79 males and 21 females) in Zhejiang Province. All recruited subjects were psychologically assessed by the school’s mental health center. A total of 62 participants (48 males and 14 females) met the inclusion and exclusion criteria of this study. Psychological evaluation physicians divided all subjects into the high-risk depression group (HD group, n = 30, “n” represents sample size) and the low-risk depression group (LD group, n = 32) according to inclusion and exclusion criteria.

The HD group was recruited according to the following criteria: (1) they had a PHQ-9 score > 9; (2) they had normal visual and audio comprehension skills. Meanwhile, the LD group was recruited according to the following criteria: (1) they had no history of mental illness or family history of mental illness; (2) they had a PHQ-9 score < 5; and (3) they had normal visual and audio comprehension skills.

In addition, exclusion criteria included (1) suffering from other mental disorders; (2) suffering from severe organic diseases; (3) having taken psychotropic drugs or received antidepressant treatment in the last two months; (4) having an intense suicidal intention; (5) having sensory development disorders (such as audio-visual problems), high myopia, hearing abnormalities, color blindness, and other eye diseases affecting visual acuity.

The study was approved by the ethics committee of Zhejiang Chinese Medical University (Approval Number: 20230627-1). All participants volunteered to participate in the experiment and signed informed consent prior to the experiment.

In the course of this trial, three people were unable to cooperate with the data collection of the whole trial due to infection with the novel coronavirus (one person in the HD group and two people in the LD group), and one person in the LD group lost data due to an abnormal collection of digital biomarker data. The above four people were excluded because of incomplete data, and the final effective sample size was 58, including 29 subjects at high risk for depression and 29 subjects at low risk for depression. The flow chart of subject screening is shown in Figure 1.

### 2.2. Design of the Paradigm and Experimental Process

Based on the hypothesis that the “emotional perception of eye movement” could characterize defects in emotional perception for the recognition, processing, and regulation of young people at high risk for depression, we designed an “eye movement emotional perception evaluation paradigm”. This paradigm consisted of four free viewing tasks: task 1 was the “European subtask of facial feature perception”, task 2 was the “European subtask of facial emotional perception”, task 3 was the “Asian subtask of facial feature perception”, task 4 was the “Asian subtask of facial emotional perception”; each task contained photos of neutral, sad, happy, and angry emotions. The stimulation materials used in tasks 1 and 2 were selected from the FACES facial emotion database, and the stimulation materials used in tasks 3 and 4 were Asian youth photos.

All participants were tested in a quiet room. We placed a comfortable and stable highchair 90 cm in front of the monitor. The viewing distance of the participants in a sitting posture was 70 ± 10 cm. We informed participants that the starting position of the eyes needed to be calibrated between each task and each face photo/face photo matrix, and participants needed to look at the gray cross on the white background for 1 s before the face photo/face photo matrix appeared. The task flow chart is shown in Figure 2. Participants were familiar with the paradigm process and manipulation methods before the formal start of the “eye movement emotional perception evaluation paradigm”. Researchers used the following prompt to all participants: “all you have to do is sit in a chair in front of the monitor and feel free to look at emotional photos that automatically appear on the monitor. The gray cross that flashes during each photo switch is eye movement calibration. The next photo will appear only when your eyes fall on the gray cross. Please feel free to look at all emotional photos that appear until the end of the paradigm.” In addition, SDS is a common self-rating depression scale, which is an effective and sensitive method of evaluating the clinical severity of depressed patients [25]. In this study, all participants completed the SDS scale prior to the “eye movement emotional perception evaluation paradigm”. The above evaluations were carried out on the same day to ensure the stability of the participants’ state of anxiety and to better clarify the effectiveness of this screening technique.

After the formal start of the “eye movement emotional perception evaluation paradigm” began, the tasks were performed sequentially. Task 1 showed single-faced photos of middle-aged European males and females in the order of neutral, sad, happy, and angry faces. After each photo appeared, the participants were free to watch it for 5 s with a total of 8 photos. Task 2 showed the face photo matrices of young males, old males, young females, and old females in the form of a two-by-two European face photo matrix (neutral on the upper left, angry on the upper right, sad on the lower left, happy on the lower right). Participants were free to watch for 10 s with a total of 4 photo matrices. Task 3 showed single-faced photos of three types of young Asian males in the order of neutrality, sadness, happiness, and anger. After each photo appeared, the participants were free to watch it for 5 s with a total of 12 photos. Task 4 showed face photo matrices of three different young Asian males in the form of a four-by-four face photo matrix. Participants were free to watch for 10 s with a total of 4 photos. The schematic diagrams of each task are shown in Figure 3.

In the paradigm process, we collected objective data from the participants’ entire process through an eye tracker and recorded them using an HCI system. Figure 4 shows partial eye movement heat maps of one low-risk and one high-risk subject with depression. The closer the position color in the heat maps was to red, the more the subject’s eye movement information was focused there, while yellow and green represented the areas with less attention. The upper parts of Figure 4a, c are eye movement heat maps of the “facial feature perception task” for the subject at low risk of depression, and the lower parts are eye movement heat maps of the “facial feature perception task” for the subject at high risk of depression. The left side of Figure 4b, d contains eye movement heat maps of the “facial emotion perception task” for the subject at low risk of depression, and the right side includes eye movement heat maps of the emotion perception task for the subject at high risk of depression.

The *m-th* face photo/face photo matrix in the *n-th* task is expressed as:(1)$$n=\left\{\begin{array}{c} \begin{array}{c} 1\;\;\;\;1\leq m\leq 8,m\in N\\ 2\;\;\;\;1\leq m\leq 4\;and\;1\leq y\leq 4,m,y\in N \end{array}\\ \begin{array}{c} 3\;\;1\leq m\leq 12,m\in N\\ 4\;\;\;\;1\leq m\leq 4\;and\;1\leq y\leq 16,m,y\in N \end{array} \end{array}\right.$$

Formula (1) means that there are a total of m face photos or face photo matrices in the *n-th* task. We recorded the *m-th* face photo of task *n-th* as *Task n_P_*_(*m*)_, and we recorded the *y-th* face photo of the *m-th* face photo matrix of task *n-th* as *Task n_P_*_(*m*)*y*_.

### 2.3. Data Acquisition

The devices used in this experiment include an Intel computer (NUC11PAHi5), a 1920 × 1080-pixel display (392 mm × 250 mm × 10 mm, 17.3 inches), and a Tobii eye tracker 5 with a sampling rate (*SR*) of 60 Hz. We constructed the “eye movement emotional perception evaluation paradigm” based on Unity and integrated it into the HCI system (including the paradigm evaluation module and data acquisition module). Original HCI data included objective evaluation data such as participants’ eye movement coordinates, the time for each task, and each photo, which was recorded in a log file (JSON format) and uploaded to the database.

### 2.4. Definition and Quantitative Analysis of Digital Biomarkers

We extracted digital biomarkers based on the objective evaluation data mentioned above.

Because tasks 1 and 3 in the “eye movement emotional perception evaluation paradigm” are “facial feature perception tasks”, the extracted digital biomarkers and analysis algorithms were similar. We extracted three types of indexes: fixation time, scan path length, and average scanning speed from the two dimensions of the gaze feature and saccade feature for face photos with different emotions. For fine-grained analysis, we divided five areas of interest (AOI) into the single-face photo, including, namely, the overall area (O), face area (F), eye-to-mouth area (EM), eye-to-nose area (EN), and eye area (E), each of which is rectangular. The generalization of digital biomarkers characterizing the “facial feature perception task” is shown in Figure 5.

The definition, abbreviation, and interpretation of digital biomarkers that characterize the “facial feature perception task” are shown in Table 1, Table 2 and Table 3.

The calculation methods of digital biomarkers that characterize the ”facial feature perception task” are as follows:

The following is an example of the calculation of digital biomarkers for an AOI area of a happy face photo:

We assumed that when participants looked at the *m-th* face photo in task *n* and the photo was a happy face, *n*
∈ {1, 3}; the range of *m* is shown in Formula (1). The eye tracker accumulated a total of *I* eye movement coordinates (*I* ∈ N) where the *i-th* eye movement coordinate was (*x_i_*, *y_i_*), (0<i ≤I, *i*
∈ N). To determine whether the *i-th* eye movement coordinate was in an AOI region (the AOI can be replaced with “O”, “F”, “EM”, “EN”, and “E” to represent a specified AOI), the formula is as follows:(2)$$f\left(i,AOI\right)=\left\{\begin{array}{c} 1,{x_{min}\leq x}_{i}\leq x_{max}\;and\;{y_{min}\leq y}_{i}\leq y_{max}\\ 0,otherwise \end{array}\right.$$
where (*x_min_*, *y_max_*) and (*x_max_*, *y_min_*) in Formula (2) are the vertex coordinates of the upper left corner and the lower right corner of the AOI region, respectively.

The *Happy*_(*FT,AOI*)_ formula is as follows:(3)$${Happy}_{\left(FT,AOI\right)}=\sum_{i=1}^{i=I}f\left(i,AOI\right)/SR\;\left(0<i\leq I,i\in N\right)$$

We set the distance between the two adjacent eye movement coordinates as *d_i_*:(4)$$d_{i}=\sqrt{{\left(x_{i+1}-x_{i}\right)}^{2}+{\left(y_{i+1}-y_{i}\right)}^{2}}\;\left(0<i\leq I-1,i\in N\right)$$

Because the eye movement tracks in each AOI were not continuous during the task, we calculated each continuous eye movement track length in each region separately and then accumulated them to obtain the scan path length in a certain AOI. We assumed that the participants had *a* continuous eye movement tracks (*a* > 0, *a* ∈ N) in a certain AOI region, and there are *k* eye movement coordinates (0 < *j* ≤ *a*, and *j*, *k* ∈ N) in the *j-th* continuous eye movement track; then, the distance of the *j-th* (*distance_j_*) continuous eye movement tracks can be obtained as follows:(5)$${distance}_{j}=\sum_{i=1}^{i=k-1}d_{i}\;\left(0<i\leq k-1,i\in N\right)$$

Hence, *Happy*_(*SPL,AOI*)_ is calculated as follows:(6)$${Happy}_{\left(SPL,AOI\right)}=\sum_{j=1}^{j=a}{distance}_{j}$$

We calculated the velocity of the eye movement track based on the length of each continuous eye movement track and then obtained the average saccade velocity in a certain area. The velocity of the *j-th* (*v_j_*) continuous eye movement track is as follows:(7)$$v_{j}=\frac{{SR\;*\;distance}_{j}}{k}$$

Then, *Happy*_(*ASS,AOI*)_ is calculated as follows:(8)$${Happy}_{\left(ASS,AOI\right)}=\sum_{j=1}^{j=a}v_{j}/a$$

Because tasks 2 and 4 in the “eye movement emotional perception evaluation paradigm” are “facial emotional perception tasks”, the extracted digital biomarkers and analysis algorithms were similar. We extracted four types of indexes for different emotions: fixation time, attention level, attention shift, and attention times from the two dimensions of gaze feature and scan feature. The generalization of digital biomarkers characterizing the “facial emotional perception task” is shown in Figure 6.

The definition, abbreviation, and interpretation of digital biomarkers characterizing the “facial emotional perception task” are shown in Table 4, Table 5, Table 6 and Table 7.

The calculation methods of digital biomarkers that characterize the “facial emotional perception task” are as follows:

During the “facial emotional perception task”, we set the time for participants to look at the *m-th* face photo matrix as *T*_(*n,m*)_ and the time for staring at the *y-th* face photo as *T*_(*n,m*)*y*_. If the photo was neutral, the *Neutral_FT_* of this photo was *T*_(*n,m*)*y*_. The *Sad_FT_*, *Happy_FT_*, and *Angry_FT_* of this photo were all 0. The calculation method of fixation time for other emotional face photos was the same. The *Neutral_AL_* of the photo could be calculated as follows:(9)$${{Task\;n}_{p\left(m\right)y}\;Neutral}_{AL}=T_{(n,m)y}/T_{\left(n,m\right)}$$

*Task n_P_*_(*m*)*y*_ in Formula (9) represents the *y-th* emotional photo of the *m-th* emotional photo matrix of task *n*. *n*
∈ {2, 4}, where the range of *m* and *y* is shown in Formula (1), and the *Sad_AL_*, *Happy_AL_*, and *Angry_AL_* of this photo are all 0. The calculation of attention level for other emotional face photos is the same.

We set the time for participants to look at the sad, angry, happy, and neutral faces in the photo matrix as *T*_(*n,m*)*_sad*_, *T*_(*n,m*)*_angry*_, *T*_(*n,m*)*_happy*_, and *T*_(*n,m*)*_neutral*_, respectively. The *Happy_AL_* calculation formula is as follows:(10)$${Happy}_{AL}=T_{(n,m)\_sad}/T_{\left(n,m\right)}$$

*Sad_AL_, Angry_AL_ and Neutral_AL_* are calculated in the same way.

The *Sad_AS_* calculation formula is as follows:(11)$${Sad}_{AS}={Sad}_{AL}-{Neutral}_{AL}$$

The *Happy_AS_* calculation formula is as follows:(12)$${Happy}_{AS}={Happy}_{AL}-{Neutral}_{AL}$$

We set the number of times the participants switched their eyes from happy face photos to non-happy face photos and then returned to happy face photos during the evaluation process as *count_happy_*. The *Happy_AT_* calculation formula is as follows:(13)$${Happy}_{AT}={count}_{happy}$$

*Neutral_AT_*, *Sad_AT_,* and *Angry_AT_* are calculated in the same way.

### 2.5. Statistical Analysis

All statistical analyses were performed using the general data analysis software SPSS 25.0. Levene’s test was used to check the homogeneity of variance. An independent sample t-test and χ^2^ test were used to compare the differences between groups for continuous variables and categorical variables, respectively. The calculation of the effect size referred to Cohen’s suggestion [26]. We used a stepwise binary logistic regression to screen digital biomarkers and investigated the contribution of digital eye movement biomarkers to depression detection. Finally, we drew the receiver operating characteristic (ROC) curve and the area under the curve to compare the performance of a single digital biomarker for the early warning of HD. For the combined early warning of multiple digital biomarkers, we used a binary logistic regression model for multivariate analysis. The value of *p* < 0.05 was considered statistically significant. Benjamini Hochberg was used to control the false discovery rate (FDR).

## 3. Results

### 3.1. Demographic and Clinical Characteristics

Subjects included 58 first-year undergraduate students aged 18 to 22. They were divided into the HD group (n = 29) and the LD group (n = 29). The participants’ demographic data and the results of the difference analysis in PHQ-9 and SDS scores between HD and LD groups are shown in Table 8. Overall, there were no significant differences in age and sex between the two groups (*p* > 0.05), but PHQ-9 and SDS scores in the HD group were significantly higher than those in the LD group (*p* < 0.001), as shown in Figure 7.

### 3.2. Eye-Tracking Data

Among the four free viewing tasks of the “eye movement emotional perception evaluation paradigm” designed in this study, HD and LD groups showed certain intergroup differences in the digital biomarkers characterizing the “facial feature perception task” and the “facial emotional perception task” (*p* < 0.05).

#### 3.2.1. Analysis of Digital Biomarkers in the “European Subtask of Facial Feature Perception”

The results of the analysis of digital biomarkers in the “European facial feature perception subtask” are shown in Table 9.

In the digital biomarkers of fixation time that characterize the “facial feature perception task”, the HD group had a longer fixation time for the overall area on the sad face photo (*Task 1_P_*_(*2*)_ *Sad*_(*FT,O*)_, *t* = −3.518, *p* = 0.001) than the LD group.

In the digital biomarkers of scan path length that characterize the “facial feature perception task”, the HD group had a shorter scan path length in the overall area on the neutral face photo (*Task 1_P_*_(*1*)_ *Neutral*_(*SPL,O*)_, *t* = 2.243, *p* = 0.029) than the LD group.

In the digital biomarkers of the scan path length that characterize the “facial feature perception task”, when scanning the neutral face photo, the HD group had a slower average scanning speed in the overall area (*Task 1_P_*_(*1*)_ *Neutral*_(*ASS,O*)_, *t* = 2.040, *p* = 0.046) and face area (*Task 1_P_*_(*1*)_ *Neutral*_(*ASS,F*)_, *t* = 2.104, *p* = 0.042) than the LD group. When scanning sad face photos, the HD group had a slower average scanning speed in the overall area (*Task 1_P_*_(*6*)_ *Sad*_(*ASS,O*)_, *t* = 2.066, *p* = 0.045) than the LD group, but had a faster average scanning speed of the eye area (*Task 1_P_*_(*2*)_ *Sad*_(*ASS,E*)_, *t* = −2.077, *p* = 0.042). The HD group had a faster average scanning speed of the eye area on angry face photos (*Task 1_P_*_(*8*)_ *Angry*_(*ASS,E*)_, *t* = −2.332, *p* = 0.023) than the LD group.

#### 3.2.2. Analysis of Digital Biomarkers in the “European Subtask of Facial Emotional Perception”

The results of the analysis of digital biomarkers in the “European subtask of facial emotional perception” are shown in Table 10.

In the digital biomarkers of attention times and attention level that characterize the “facial emotional perception task”, attention times (*Task 2_P_*_(*4*)_ *Neutral_AT_*, *t* = 2.383, *p* = 0.021) and attention level (*Task 2_P_*_(*4*)_ *Neutral_AL_*, *t* = 2.002, *p* = 0.049) to the neutral face photo in the HD group were lower than the LD group.

#### 3.2.3. Analysis of Digital Biomarkers in the “Asian Subtask of Facial Feature Perception”

The results of the analysis of digital biomarkers in the “Asian subtask of facial feature perception” are shown in Table 11.

In the digital biomarkers of scan path length that characterize the “facial feature perception task”, when scanning neutral face photos, the HD group had a shorter scan path length in the overall area (*Task 3_P_*_(*5*)_ *Neutral*_(*SPL,O*)_, *t* = 2.183, *p* = 0.033), face area (*Task 3_P_*_(*5*)_ *Neutral*_(*SPL,F*)_, *t* = 2.382, *p* = 0.021; *Task 3_P_*_(*9*)_ *Neutral*_(*SPL,F*)_, *t* = 2.138, *p* = 0.038) and eye-to-mouth area (*Task 3_P_*_(*5*)_ *Neutral*_(*SPL,EM*)_, t = 2.083, *p* = 0.042; *Task 3_P_*_(*9*)_
*Neutral*_(*SPL,EM*)_, *t* = 2.153, *p* = 0.036) than the LD group. When scanning the angry face photo, the HD group had a shorter scan path length in the face area (*Task 3_P_*_(*8*)_ *Angry*_(*SPL,O*)_, *t* = 2.133, *p* = 0.037) than the LD group. And when scanning the happy face photo, the HD group had a shorter scan path length in the face area (*Task 3_P_*_(*11*)_ *Happy*_(*SPL,F*)_, *t* = 2.310, *p* = 0.026) and eye-to-mouth area (*Task 3_P_*_(*11*)_ *Happy*_(*SPL,EM*)_, *t* = 2.814, *p* = 0.007) than the LD group.

In the digital biomarkers of average scanning speed that characterize the “facial feature perception task”, when scanning the neutral face photo, the HD group had a shorter average scanning speed in the face area (*Task 3_P_*_(*5*)_ *Neutral*_(*ASS,F*)_, *t* = 2.127, *p* = 0.038) and eye-to-mouth area (*Task 3_P_*_(*5*)_ *Neutral*_(*ASS,EM*)_, *t* = 3.054, *p* = 0.003) than the LD group. And when scanning sad face photos, the HD group had a shorter average scanning speed of the face area (*Task 3_P_*_(*6*)_ *Neutral*_(*ASS,F*)_, *t* = 2.188, *p* = 0.033) than the LD group.

#### 3.2.4. Analysis of Digital Biomarkers in the “Asian Subtask of Facial Emotional Perception”

The results of the analysis of digital biomarkers in the “Asian subtask of facial emotional perception” are shown in Table 12.

In the digital biomarkers of fixation time that characterize the “facial emotional perception task”, the HD group had a shorter fixation time for happy face photos (*Task 4_P_*_(*2*)*16*_ *Happy_FT_*, *t* = 2.250, *p* = 0.028; *Task 4_P_*_(*4*)*12*_ *Happy_FT_*, *t* = 2.833, *p* = 0.006) than the LD group; and had a longer fixation time for sad face photo (*Task 4_P_*_(*4*)*3*_ *Sad_FT_*, *t* = −3.023, *p* = 0.004) than the LD group.

In the digital biomarkers of the attention level that characterize the “facial emotional perception task”, the attention level to the neutral face photo (*Task 4_P_*_(*4*)_ *Neutral_AL_*, *t* = 2.292, *p* = 0.026), and happy face photos (*Task 4_P_*_(*2*)*16*_ *Happy_AL_*, *t* = 2.279, *p* = 0.026; *Task 4_P_*_(*4*)*12*_
*Happy_AL_*, *t* = 2.283, *p* = 0.007) in the HD group were lower than the LD group; but the attention level to the sad face photo (*Task 4_P_*_(*4*)*3*_
*Sad_AL_*, *t* = −3.042, *p* = 0.004) in the HD group was higher than the LD group.

### 3.3. Binary Logistic Regression

Considering that the sample size of this study was small and too many variables could lead to overfitting in the model, this study used stepwise regression analysis to reduce the dimensions of eye movement digital biomarkers. We took all the above-mentioned eye movement digital biomarkers with intergroup differences (*p* < 0.05) as candidate variables and finally selected a total of seven eye movement digital biomarkers for the logical model with forward selection. These digital biomarkers included the following: *Task 1_P_*_(*2*)_ *Sad*_(*FT,O*)_, *Task 1_P_*_(*1*)_ *Neutral*_(*ASS,F*)_, *Task 2_P_*_(*4*)*1*_ *Neutral_AL_*, *Task 3_P_*_(*5*)_ *Neutral*_(*ASS,EM*)_, *Task 4_P_*_(*4*)*12*_ *Happy_FT_*, *Task 4_P_*_(*4*)*3*_ *Sad_AL_*, *Task 4_P_*_(*4*)_
*Neutral_AL_*. Intergroup differences in these digital biomarkers between the HD and LD groups are shown in Figure 8.

### 3.4. The Recognition of Depression

We plotted the ROC curve to evaluate the predictive capacity of screened eye movement digital biomarkers for the high-risk population of depression, and the area under the curve (AUC) of a single eye movement digital biomarker reached 0.650, as shown in Figure 9.

Then, we plotted the ROC curve for seven digital biomarkers combined to warn young people at high risk of depression, and the AUC reached 0.974. Its early warning effectiveness exceeded that of SDS, a clinically and commonly used diagnostic scale (AUC = 0.936), as shown in Figure 10. The results demonstrate that the clinical efficacy of this novel intelligent HCI screening method is based on the “eye movement emotional perception evaluation paradigm”.

### 3.5. Eye Movement Data for Visualization Heat Maps

In this study, the eye tracker was used to collect dynamic and objective eye movement data from all subjects during the “facial feature perception task” and “facial emotion perception task”. Based on the above selected digital biomarkers, we mapped the visual eye movement heat maps of HD and LD groups to the corresponding emotional facial images, as shown in Figure 11, Figure 12 and Figure 13. Each eye movement heat map included eye movement data from all participants in HD or LD groups.

In the “European subtask of facial feature perception” and “Asian subtask of facial feature perception” of the “facial feature perception task”, the eye movement fixation range in the HD group was larger than the LD group, as shown in Figure 11a–c.

In the “European subtask of facial emotional perception” of the “facial emotion perception task”, the eye movement information of the HD group was more concentrated on negative emotional photos such as anger (Figure 12(a)-②) and sadness (Figure 12(a)-③) than in the LD group.

In the “Asian subtask of facial emotional perception” of the “facial emotion perception task”, the eye movement information of the HD group was more concentrated on negative emotional photos such as anger (Figure 13(a)-②) and sadness (Figure 13(a)-③) than that the LD group.

## 4. Discussion

In this paper, we propose a novel, intelligent, and rapid screening method for depression risk in young people based on eye-tracking technology. It has the advantage of short, time-consuming, cost-effective, and anti-antagonism properties while eliminating the need for specialized medical personnel to operate the task. We designed the “eye movement emotional perception evaluation paradigm”, including the “facial feature perception task” and “facial emotion perception task”, and extracted eye movement digital biomarkers from objective evaluation data. Based on the above digital biomarkers, we found that young people at high risk of depression exhibited different “saccade” and “fixation” characteristics of eye movement under different emotional and facial stimuli compared to those at low risk of depression. We ultimately selected seven digital biomarkers that represented emotional perception, recognition, processing, and regulation defects in young people at high risk of depression, and the combined early warning efficiency reached 0.974.

Our experimental results show that in both the “facial feature perception task” and the “facial emotion perception task”, the fixation time for sad faces in the HD group was longer than that in the LD group. In the “facial emotional perception task”, the attention level of sad faces in the HD group was significantly higher than those in the LD group, while the fixation time and attention level of happy faces and neutral faces in the HD group were lower than those in the LD group. The above results are consistent with Lazarov et al.’s findings compared to other emotional stimuli, where people with depression are more likely to focus on negative emotional stimuli. Current neuropathological studies have found significant functional abnormalities in the amygdala in patients with depression [27]. The emotional center of the brain, the amygdala, plays a critical role in emotional perception, recognition, processing, and regulation [28,29,30]. Attention bias toward negative emotions in young people at high risk of depression may be due to the overactivation of the amygdala in the brain, resulting in an abnormal emotional stimulation response function [31]. Therefore, people in the HD group may have significant changes in fixation time and attention levels to different emotional faces in the process of performing the “facial feature perception task” and “facial emotion perception task” due to the overactivation of the amygdala. It is worth noting that the difference in the fixation of positive and negative emotional faces in the “facial emotional perception task” in the HD group appeared only in Asian faces, while no such differences were noted in European faces. This suggests that facial expressions across cultures and ethnicities may affect fixation stability.

We also found that in the “facial feature perception task”, the HD group shortened the scanning path length of various emotional photos and decreased the average scanning speed, which was consistent with Takahashi et al. A short scan path length and reduced average scanning speed in the HD group could be associated with psychomotor retardation (PMR). PMR likely alters various behaviors in individuals, including motor skills, psychological activities, and speech. Its main manifestations include delayed responses, sluggish movements, and reduced fluency and coordination of actions. Previous research has shown significant functional impairments within the frontal cortex of patients with depression [32]. The frontal cortex is involved in a multitude of behavioral and cognitive processes, including cognitive control, attention, executive functions, and emotional regulation, and collaborates with the amygdala in regulating emotions and behaviors [33,34,35]. The symptoms of PMR observed in high-risk youths for depression may be associated with deficiencies in the functioning of the frontal cortex and abnormalities in dopamine neurotransmission, causing motor impairments [36]. After narrowing the AOI, we observed an increased average scanning speed of the eye region focused on sad and angry facial expressions in the HD group. This finding contrasts with the overall viewing pattern of emotional photographs in the HD group, suggesting a potential tendency among high-risk youths for depression to avoid prominent facial features when recognizing facial emotions. Carmel et al. proposed that avoidance of prominent facial features may be an episodic-cognitive trait in individuals with affective disorders closely related to negative mood states. This avoidance of significant facial features may occur due to the dysfunctional emotional processing caused by abnormalities in the amygdala and frontal cortex function in individuals with affective disorders. As a result, the HD group may exhibit slow eye movements and an inclination to avoid the facial area during the “facial feature perception task”, potentially due to abnormalities in the frontal cortex and amygdala function.

We used eye movement heat maps to perform a more intuitive visual analysis of the results of the “facial feature perception task” and “facial emotion perception task” between the HD and LD groups. This analysis further explored attention bias and attention deficits in these groups. The results from eye movement heat maps revealed that the HD group exhibited longer gaze durations on negative emotional images, which was indicated by more intense red colors in the heat maps. This indicates that eye-tracking information in the HD group is more focused on negative emotional images, providing additional confirmation of a negative attention bias in the HD group. The results of eye movement heat maps also indicate that the range of eye movement activity in the HD group is larger than that in the LD group, with more scattered gaze points. This suggests a reduced level of attention focus among high-risk youths for depression, which is consistent with Watts et al.’s findings. This distraction may be due to the attention deficit caused by an impaired prefrontal cortex function in young people at high risk for depression [37]. Therefore, it is plausible that the HD group shows a lack of concentration during the execution of the “facial feature perception task” and “facial emotion perception task” due to impaired frontal cortex function. Additionally, they showed an increased range of eye movement activity.

There are also some limitations to this study. First, our total sample size is limited, and we need to further increase the sample size for future experiments to expand the universality of these research results. Second, because all participants in this study had never experienced psychotherapy and counseling intervention, protective factors that could prevent or relieve anxiety symptoms were not evaluated in this study. In the future, we aim to continue to explore the early warning performance of our new method for people who have received psychotherapy and counseling intervention. Third, in this study, we did not conduct a further and detailed study on the specific neural mechanisms that may be involved in the results of the “facial feature perception task” and the “facial emotion perception task.” In the future, these follow-up mechanisms should be further explored by molecular or functional imaging technology exemplified by positron emission tomography-computed tomography (PET-CT) and functional nuclear magnetic imaging. In addition, in the aspect of data analysis, we adopted the stepwise method of binary logistic regression. This method optimizes the model by adding or deleting features step by step to improve the prediction accuracy. Although this method is widely used in the field of disease screening, it is prone to over-fitting when the number of samples is less than the number of potential predictive variables. In addition, the digital biomarkers selected by the binary logistic regression step-by-step method may also change with different data subsets, leading to the poor generalization effect of the early warning model. In the clinical application of digital biomarkers, because this study is interdisciplinary, it involves the experience of clinicians in the diagnosis and treatment, psychologists’ paradigm design, and so on. However, due to the limited funds in this study, we have not scored the existing digital biomarkers based on the Analytic Hierarchy Process (AHP) combined with the opinions of experts in various fields. Therefore, there may be two shortcomings in this paper: firstly, the sensitivity, specificity, and clinical application goal have not been fully considered in the selection of the ROC threshold in this paper. Secondly, based on the actual clinical application, this paper has not given personalized early warning results for different populations and environmental characteristics.

## 5. Conclusions

To sum up, we proposed a hypothesis that “emotional perception of eye movement” can characterize defects in emotional perception, recognition, processing, and regulation in young people at high risk of depression. Based on this hypothesis, we propose a new intelligent and rapid screening method for depression risk in young people based on eye tracking technology and design the “eye movement emotional perception evaluation paradigm”. This method can intelligently extract and analyze digital multi-dimensional eye movement biomarkers in the process of subjects implementing the “eye movement emotional perception evaluation paradigm”. Finally, the rapid screening of young people at high risk for depression was achieved. After clinical verification, this method is suitable for screening high-risk individuals with depression between the ages of 18 and 22, and its early warning effectiveness can reach 0.974. The new intelligent and rapid screening method for depression risk in young people based on eye tracking technology proposed in this paper has the advantages of high speed, low costs, high early warning efficiency, and high intelligence levels. It provides a new way for young people at high risk of depression to have early large-scale screening for depression without doctors. In the future, we aim to further expand the sample size to include participants of different ages and anxiety levels to expand the universality of these research results. Additionally, we also aim to combine medical imaging technology to further explore the specific neural mechanism involved in this new method.

## Figures and Tables

**Figure 1 brainsci-13-01415-f001:**
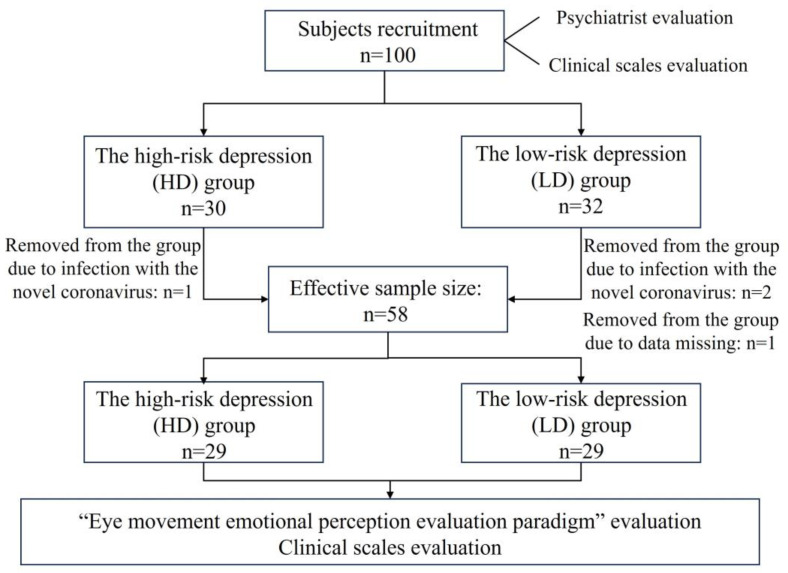
The flow chart of subject screening.

**Figure 2 brainsci-13-01415-f002:**
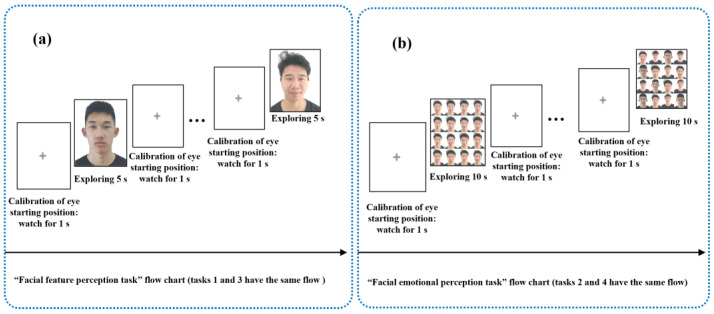
The task flow charts. (**a**) “Facial feature perception task” flow. (**b**) “Facial emotional perception task” flow.

**Figure 3 brainsci-13-01415-f003:**
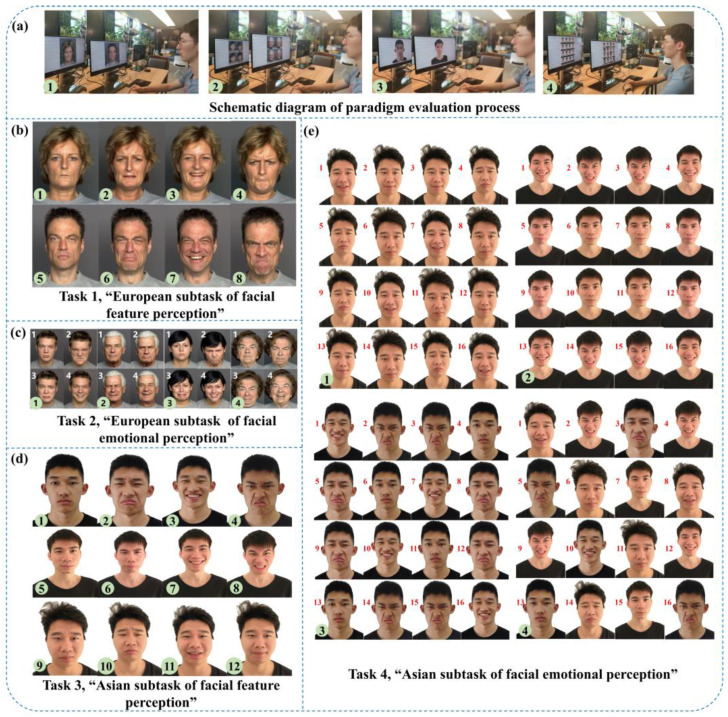
The schematic diagrams of each task. The sequence number in the green circle represents the order in which the pictures appeared. (**a**) Schematic diagram of the paradigm evaluation process. (**b**) Task 1, “European subtask of facial feature perception”; (**c**) Task 2, “European subtask of facial emotional perception”; the white number represents the number of each sub-photo in the face photo matrix. (**d**) Task 3, “Asian subtask of facial feature perception”; (**e**) Task 4, “Asian subtask of facial emotional perception”; the red number represents the number of each sub-photo in the face photo matrix.

**Figure 4 brainsci-13-01415-f004:**
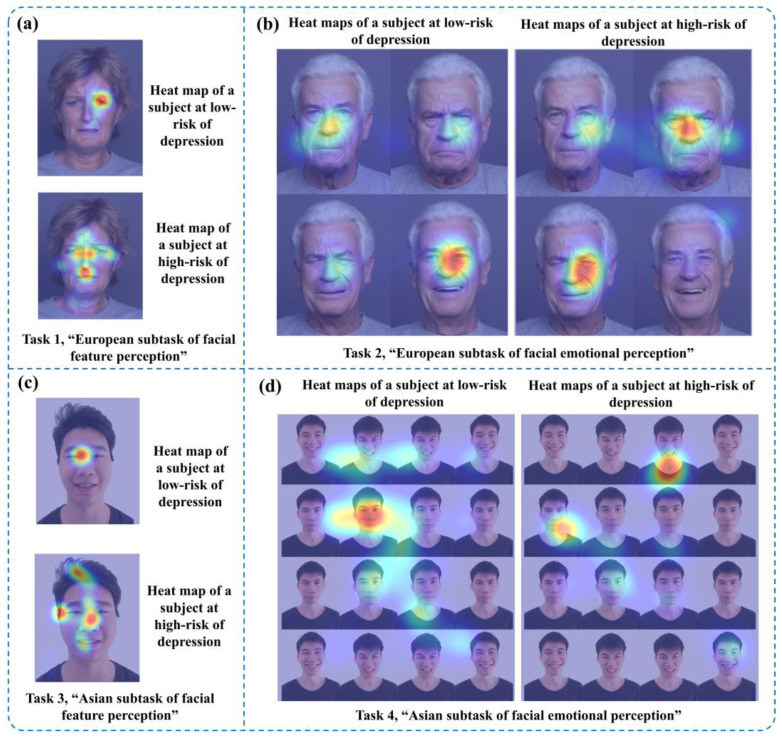
Comparison of partial eye movement heat maps for two subjects. The closer the position color in the heat maps was to red, the more the subject’s eye movement information was focused there, while yellow and green represented the areas with less attention. (**a**,**c**) Eye movement heat maps for the “facial feature perception task”. (**b**,**d**) Eye movement heat maps for the “facial emotional perception task”.

**Figure 5 brainsci-13-01415-f005:**
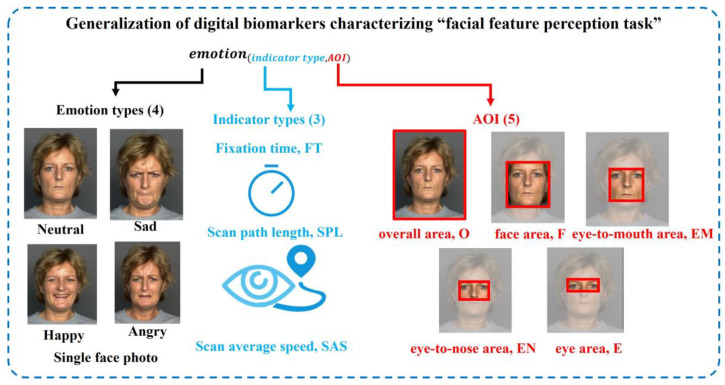
Generalization of digital biomarkers characterizing the “facial feature perception task”.

**Figure 6 brainsci-13-01415-f006:**
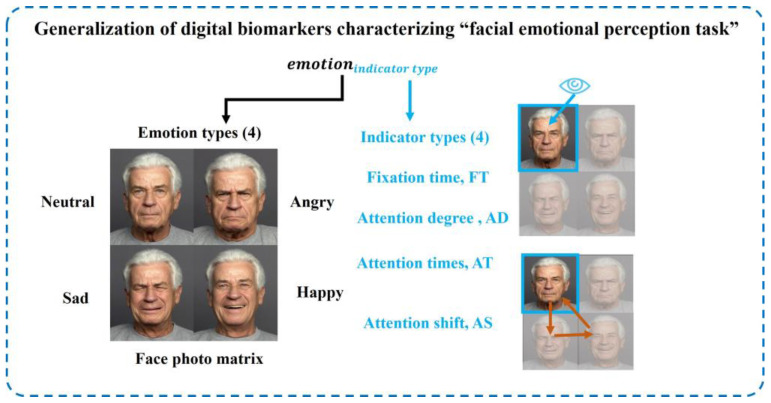
Generalization of digital biomarkers characterizing the “facial emotional perception task”.

**Figure 7 brainsci-13-01415-f007:**
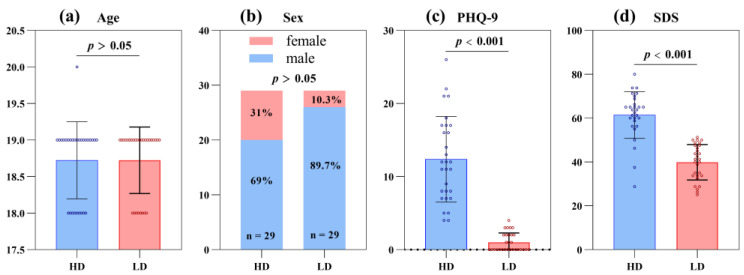
Data distribution of (**a**) Age in HD and LD groups. (**b**) Sex in HD and LD groups. (**c**) PHQ-9 score in HD and LD groups. (**d**) SDS score in HD and LD groups.

**Figure 8 brainsci-13-01415-f008:**
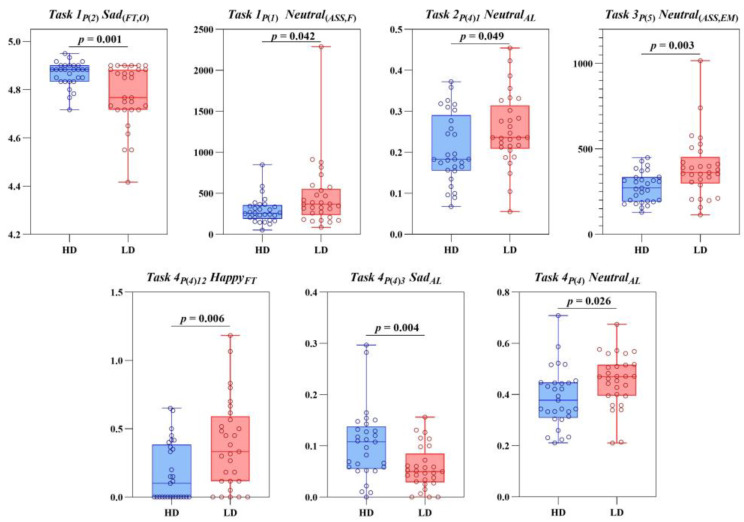
The analysis results of the differences between HD and LD groups for the seven selected eye movement digital biomarkers.

**Figure 9 brainsci-13-01415-f009:**
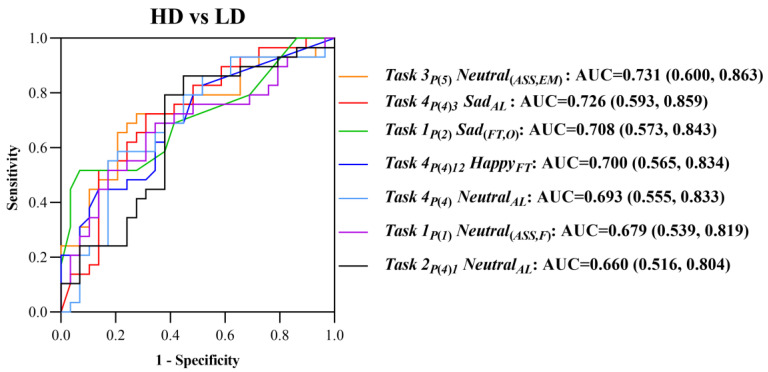
ROC curves of seven selected eye movement digital biomarkers for the early warning of depression in high-risk populations.

**Figure 10 brainsci-13-01415-f010:**
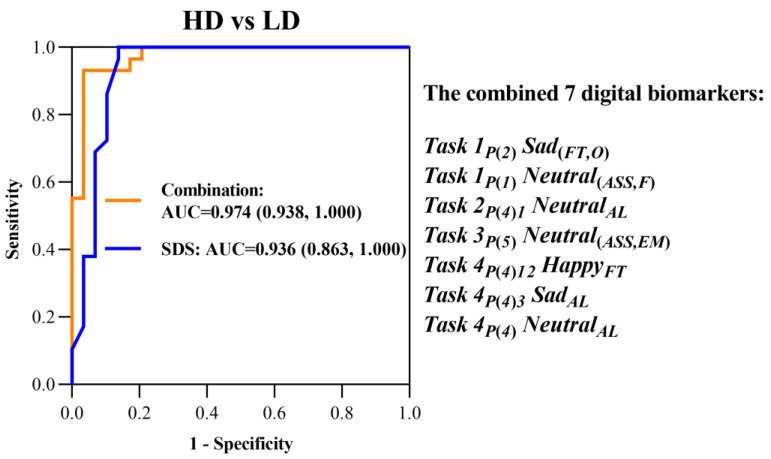
ROC curves of SDS and the combined seven eye movement digital biomarkers of HD and LD groups.

**Figure 11 brainsci-13-01415-f011:**
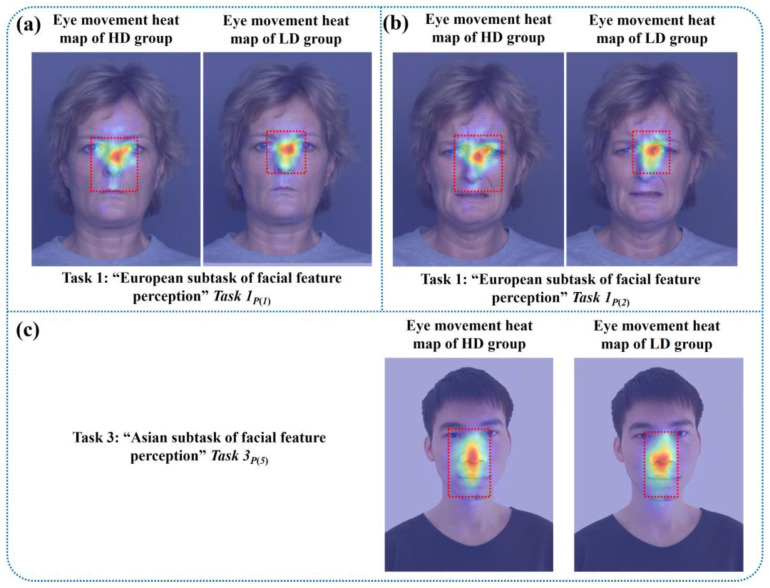
Visual eye movement heat maps of HD and LD groups in the “European subtask of facial feature perception” and “Asian subtask of facial feature perception”. (**a**) Eye movement heat maps of HD and LD groups in task 1, picture 1. (**b**) Eye movement heat maps of HD and LD groups in task 1, picture 2. (**c**) Eye movement heat maps of HD and LD groups in task 3, picture 5.

**Figure 12 brainsci-13-01415-f012:**
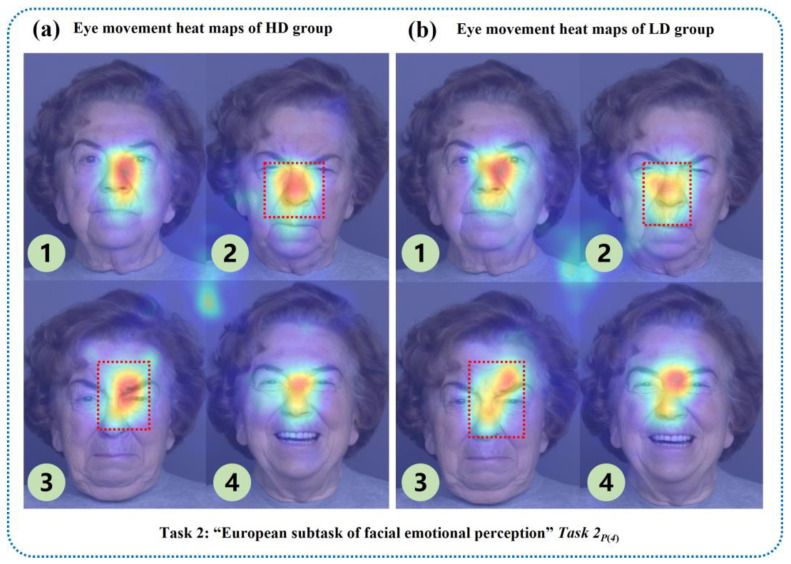
Visual eye movement heat maps of HD and LD groups in the “European subtask of facial emotional perception”.

**Figure 13 brainsci-13-01415-f013:**
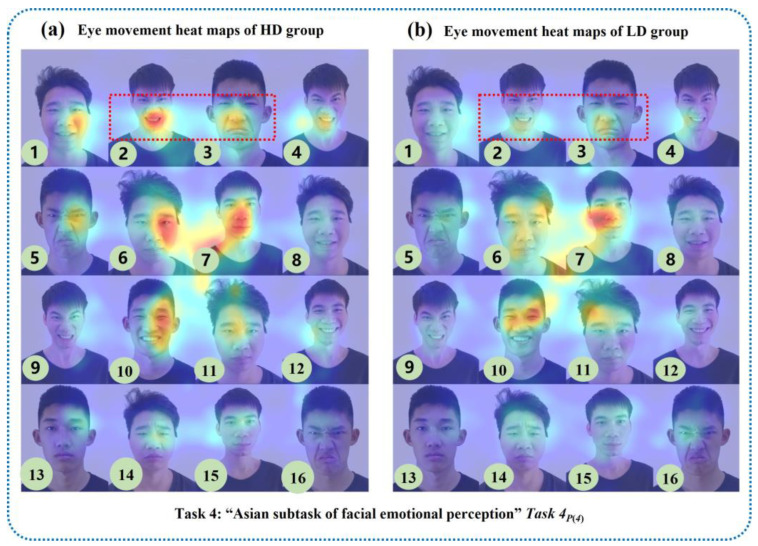
Visual eye movement heat maps of HD and LD groups in the “Asian subtask of facial emotional perception”.

**Table 1 brainsci-13-01415-t001:** Digital biomarkers characterizing the “facial feature perception task” (fixation time).

Digital Biomarker	Abbreviation	Unit	Interpretation
Fixation time of the neutral face—AOI	*Neutral* _(*FT,AOI*)_	s	Means the amount of time participants spent looking at the AOI area of a neutral face photo during the task.
Fixation time of the sad face—AOI	*Sad* _(*FT,AOI*)_	s	Means the amount of time participants spent looking at the AOI area of a sad face photo during the task.
Fixation time of the happy face—AOI	*Happy* _(*FT,AOI*)_	s	Means the amount of time participants spent looking at the AOI area of a happy face photo during the task.
Fixation time of the angry face—AOI	*Angry* _(*FT,AOI*)_	s	Means the amount of time participants spent looking at the AOI area of an angry face photo during the task.

Note: (AOI can be replaced with “O”, “F”, “EM”, “EN”, and “E” to represent a specified AOI), the fixation time for the neutral face—F of the *m-th* face photo in task 1 may be abbreviated as *Task 1_P_*_(*m*)_
*Neutral*_(*FT,F*)_.

**Table 2 brainsci-13-01415-t002:** Digital biomarkers characterizing the “facial feature perception task” (scan path length).

Digital Biomarker	Abbreviation	Unit	Interpretation
Scan path length of the neutral face—AOI	*Neutral* _(*SPL,AOI*)_	px	Means the length of the scan path participants looked at in the AOI area of a neutral face photo during the task.
Scan path length of the sad face—AOI	*Sad* _(*SPL,AOI*)_	px	Means the length of the scan path participants looked at in the AOI area of a sad face photo during the task.
Scan path length of the happy face—AOI	*Happy* _(*SPLAOI*)_	px	Means the length of the scan path participants looked at in the AOI area of a happy face photo during the task.
Scan path length of the angry face—AOI	*Angry* _(*SPL,AOI*)_	px	Means the length of the scan path participants looked at in the AOI area of an angry face photo during the task.

Note: The scan path length of the neutral face—F of the *m-th* face photo in task 1 may be abbreviated as *Task 1_P_*_(*m*)_
*Neutral*_(*SPL,F*)_.

**Table 3 brainsci-13-01415-t003:** Digital biomarkers characterizing the “facial feature perception task” (average scanning speed).

Digital Biomarker	Abbreviation	Unit	Interpretation
Average scanning speed of the neutral face—AOI	*Neutral* _(*ASS,AOI*)_	px/s	Means the average scan speed at which participants looked at the AOI area of a neutral photo during the task.
Average scanning speed of the sad face—AOI	*Sad* _(*ASS,AOI*)_	px/s	Means the average scan speed at which participants looked at the AOI area of a sad face photo during the task.
Average scanning speed of the happy face—AOI	*Happy* _(*ASS,AOI*)_	px/s	Means the average scan speed at which participants looked at the AOI area of a happy face photo during the task.
Average scanning speed of the angry face—AOI	*Angry* _(*ASS,AOI*)_	px/s	Means the average scan speed at which participants looked at the AOI area of an angry face photo during the task.

Note: The average scanning speed of the neutral face—F of the *m-th* face photo in task 1 may be abbreviated as *Task 1_P_*_(*m*)_
*Neutral*_(*ASS,F*)_.

**Table 4 brainsci-13-01415-t004:** Digital biomarkers characterizing the “facial emotional perception task” (fixation time).

Digital Biomarker	Abbreviation	Unit	Interpretation
Fixation time of the neutral face	*Neutral_FT_*	s	Means the amount of time participants spent looking at a neutral face in the photo matrix during the task.
Fixation time of the sad face	*Sad_FT_*	s	Means the amount of time participants spent looking at a sad face in the photo matrix during the task.
Fixation time of the happy face	*Happy_FT_*	s	Means the amount of time participants spent looking at a happy face in the photo matrix during the task.
Fixation time of the angry face	*Angry_FT_*	s	Means the amount of time participants spent looking at an angry face in the photo matrix during the task.

Note: The fixation time for the happy face of the *y-th* photo in the *m-th* face photo matrix of task 4 may be abbreviated as *Task 4_P_*_(*m*)*y*_
*Happy_FT_*.

**Table 5 brainsci-13-01415-t005:** Digital biomarkers characterizing the “facial emotional perception task” (attention level).

Digital Biomarker	Abbreviation	Unit	Interpretation
Attention level of the neutral face	*Neutral_AL_*	%	Means the ratio of the amount of time participants spent staring at neutral face photos during the task compared to the amount of time they spent looking at the photo matrix.
Attention level of the sad face	*Sad_AL_*	%	Means the ratio of the amount of time participants spent staring at sad face photos during the task compared to the amount of time they spent looking at the photo matrix.
Attention level of the happy face	*Happy_AL_*	%	Means the ratio of the amount of time participants spent staring at happy face photos during the task compared to the amount of time they spent looking at the photo matrix.
Attention level of the angry face	*Angry_AL_*	%	Means the ratio of the amount of time participants spent staring at angry face photos during the task compared to the amount of time they spent looking at the photo matrix.

Note: The attention level of the happy face of the *y-th* photo in the *m-th* face photo matrix of task 4 may be abbreviated as *Task 4_P_*_(*m*)*y*_
*Happy_AL_*; the attention level of all happy faces in the *m-th* face photo matrix of task 4 may be abbreviated as *Task 4_P_*_(*m*)_
*Happy_AL_*.

**Table 6 brainsci-13-01415-t006:** Digital biomarkers characterizing the “facial emotional perception task” (attention shift).

Digital Biomarker	Abbreviation	Unit	Interpretation
Attention shift of the sad face	*Sad_AS_*	%	Means that participants’ were *Sad_AL_* minus *Neutral_AL_* during the task.
Attention shift of the happy face	*Happy_AS_*	%	Means that participants’ were *Happy_AL_* minus *Neutral_AL_* during the task.

Note: The attention shift of all happy faces in the *m-th* face photo matrix of task 4 may be abbreviated as *Task 4_P_*_(*m*)_
*Happy_AS_*.

**Table 7 brainsci-13-01415-t007:** Digital biomarkers characterizing the “facial emotional perception task” (attention times).

Digital Biomarker	Abbreviation	Unit	Interpretation
Attention times of the neutral face	*Neutral_AT_*	time	Means the number of times participants switched back and forth during the task to look at neutral face photos in the photo matrix.
Attention times of the sad face	*Sad_AT_*	time	Means the number of times participants switched back and forth during the task to look at sad face photos in the photo matrix.
Attention times of the happy face	*Happy_AT_*	time	Means the number of times participants switched back and forth during the task to look at happy face photos in the photo matrix.
Attention times of the angry face	*Angry_AT_*	time	Means the number of times participants switched back and forth during the task to look at angry face photos in the photo matrix.

Note: The attention times of all happy faces in the *m-th* face photo matrix of task 4 may be abbreviated as *Task 4_P_*_(*m*)_
*Happy_AT_*.

**Table 8 brainsci-13-01415-t008:** Participants’ demographics and difference analysis in PHQ-9 and SDS scores between HD and LD groups.

	HD Group (*n* = 29)	LD Group (*n* = 29)	*p* Value
Age	18.72 (0.53)	18.72 (0.45)	1.000
Sex (female/male)	9/20	3/26	0.052
PHQ-9 **	12.38 (5.86)	1.00 (1.28)	<0.001
SDS **	61.42 (10.65)	39.87 (8.11)	<0.001

Note: Continuous variables are presented as [mean (SD)]. ** indicates a significant difference between the two groups, at *p* < 0.01.

**Table 9 brainsci-13-01415-t009:** The results of the analysis of digital biomarkers in the “European subtask of facial feature perception”.

	HD Group (*n* = 29)	LD Group (*n* = 29)	*t* (*p* Value)	FDR	Cohen *d*
*Task 1_P_*_(*2*)_ *Sad*_(*FT,O*)_ **	4.87 (0.05)	4.78 (0.13)	−3.518 (*p* = 0.001)	0.027	0.914
*Task 1_P_*_(*1*)_ *Neutral*_(*SPL,O*)_ *	1470.45 (720.71)	1967.80 (951.87)	2.243 (*p* = 0.029)	0.047	0.589
*Task 1_P_*_(*1*)_ *Neutral*_(*ASS,O*)_ *	326.83 (195.61)	454.61 (274.80)	2.040 (*p* = 0.046)	0.048	0.536
*Task 1_P_*_(*1*)_ *Neutral*_(*ASS,F*)_ *	293.23 (158.23)	466.23 (413.55)	2.104 (*p* = 0.042)	0.047	0.553
*Task 1_P_*_(*2*)_ *Sad*_(*ASS,E*)_ *	210.91 (148.36)	144.01 (89.81)	−2.077 (*p* = 0.042)	0.047	0.546
*Task 1_P_*_(*6*)_ *Sad*_(*ASS,O*)_ *	351.96 (198.86)	509.62 (359.65)	2.066 (*p* = 0.045)	0.048	0.543
*Task 1_P_*_(*8*)_ *Angry*_(*ASS,E*)_ *	256.32 (183.21)	165.90 (100.15)	−2.332 (*p* = 0.023)	0.047	0.612

Note: Continuous variables are presented as [mean (SD)]. * indicates a significant difference between the two groups, at *p* < 0.05; ** indicates a significant difference between the two groups, at *p* < 0.01.

**Table 10 brainsci-13-01415-t010:** The results of the analysis of digital biomarkers in the “European subtask of facial emotional perception”.

	HD Group (*n* = 29)	LD Group (*n* = 29)	*t* (*p* Value)	FDR	Cohen *d*
*Task 2_P_*_(*4*)_ *Neutral_AT_* *	2.79 (1.15)	3.69 (1.67)	2.383 (*p* = 0.021)	0.047	0.628
*Task 2_P_*_(*4*)_ *Neutral_AL_* *	0.21 (0.09)	0.25 (0.09)	2.002 (*p* = 0.049)	0.049	0.444

Note: Continuous variables are presented as [mean (SD)]. * indicates a significant difference between the two groups at *p* < 0.05.

**Table 11 brainsci-13-01415-t011:** The results of the analysis of digital biomarkers in the “Asian subtask of facial feature perception”.

	HD Group (*n* = 29)	LD Group (*n* = 29)	*t* (*p* Value)	FDR	Cohen *d*
*Task 3_P_*_(*5*)_ *Neutral*_(*SPL,O*)_ *	1992.05 (975.19)	2683.48 (1399.17)	2.183 (*p* = 0.033)	0.047	0.573
*Task 3_P_*_(*5*)_ *Neutral*_(*SPL,F*)_ *	1341.19 (512.77)	1751.21 (772.15)	2.382 (*p* = 0.021)	0.047	0.626
*Task 3_P_*_(*5*)_ *Neutral*_(*SPL,EM*)_ *	1170.81 (402.88)	1466.49 (649.83)	2.083 (*p* = 0.042)	0.047	0.547
*Task 3_P_*_(*8*)_ *Angry*_(*SPL,O*)_ *	1555.07 (778.69)	2094.67 (1117.45)	2.133 (*p* = 0.037)	0.047	0.560
*Task 3_P_*_(*9*)_ *Neutral*_(*SPL,F*)_ *	1485.62 (448.82)	1855.47 (816.40)	2.138 (*p* = 0.038)	0.047	0.561
*Task 3_P_*_(*9*)_ *Neutral*_(*SPL,EM*)_ *	1199.24 (381.03)	1464.16 (542.22)	2.153 (*p* = 0.036)	0.047	0.565
*Task 3_P_*_(*11*)_ *Happy*_(*SPL,F*)_ *	1214.08 (436.22)	1583.37 (742.26)	2.310 (*p* = 0.026)	0.047	0.607
*Task 3_P_*_(*11*)_ *Happy*_(*SPL,EM*)_ **	1019.00 (369.23)	1374.96 (572.30)	2.814 (*p* = 0.007)	0.027	0.739
*Task 3_P_*_(*5*)_ *Neutral*_(*ASS,F*)_ *	319.89 (134.10)	437.50 (265.85)	2.127 (*p* = 0.038)	0.047	0.559
*Task 3_P_*_(*5*)_ *Neutral*_(*ASS,EM*)_ **	276.07 (88.59)	390.67 (181.63)	3.054 (*p* = 0.003)	0.027	0.802
*Task 3_P_*_(*6*)_ *Neutral*_(*ASS,F*)_ *	264.69 (151.77)	371.66 (215.10)	2.188 (*p* = 0.033)	0.047	0.575

Note: Continuous variables are presented as [mean (SD)]. * indicates a significant difference between the two groups, at *p* < 0.05; ** indicates a significant difference between the two groups, at *p* < 0.01.

**Table 12 brainsci-13-01415-t012:** The results of the analysis of digital biomarkers in the “Asian subtask of facial emotional perception”.

	HD Group (*n* = 29)	LD Group (*n* = 29)	*t* (*p* Value)	FDR	Cohen *d*
*Task 4_P_*_(*2*)*16*_ *Happy_FT_* *	0.28 (0.27)	0.46 (0.35)	2.250 (*p* = 0.028)	0.047	0.576
*Task 4_P_*_(*4*)*3*_ *Sad_FT_* **	0.99 (0.66)	0.56 (0.41)	−3.023 (*p* = 0.004)	0.027	0.783
*Task 4_P_*_(*4*)*12*_ *Happy_FT_* **	0.18 (0.22)	0.39 (0.33)	2.833 (*p* = 0.006)	0.027	0.749
*Task 4_P_*_(*4*)_ *Neutral_AL_* *	0.39 (0.12)	0.45 (0.10)	2.292 (*p* = 0.026)	0.047	0.608
*Task 4_P_*_(*2*)*16*_ *Happy_AL_* *	0.03 (0.03)	0.05 (0.04)	2.279 (*p* = 0.026)	0.047	0.566
*Task 4_P_*_(*4*)*12*_ *Happy_AL_* **	0.02 (0.02)	0.04 (0.03)	2.823 (*p* = 0.007)	0.027	0.784
*Task 4_P_*_(*4*)*3*_ *Sad_AL_* **	0.10 (0.07)	0.06 (0.04)	−3.042 (*p* = 0.004)	0.027	0.702

Note: Continuous variables are presented as [mean (SD)]. * indicates a significant difference between the two groups, at *p* < 0.05; ** indicates a significant difference between the two groups, at *p* < 0.01.

## Data Availability

Data will be shared upon request.

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
