# Peer review of "Research on a New Intelligent and Rapid Screening Method for Depression Risk in Young People Based on Eye Tracking Technology"

_brainsci, 2023, doi:10.3390/brainsci13101415_

Round 1
Reviewer 1 Report
Dear Editor,
I appreciate the opportunity to review manuscript brainsci-2604621 entitled:
"Research on a New Intelligent and Rapid Screening Method for Depression Risk in Young People Based on Eye Tracking Technology"
I commend the authors for describing this critical and timely issue. The paper is interesting and well-written; however, I would like to highlight some issues that merit revision:
It is not detectable in the manuscript whether an assessment was made of what might be a protective factor through psychotherapy and counseling intervention, which people often use without prescription to prevent or alleviate symptoms at onset. I would ask the authors if this was evaluated during the experiment or in the inclusion criteria; please add a short paragraph on this issue; if data are unavailable, add them to the limitations.
Author Response
Dear reviewer,
Thank you for your valuable suggestions. We have revised the article in response to your suggestions. The following is the response to each proposal and the location of the changes:
Comments 1: It is not detectable in the manuscript whether an assessment was made of what might be a protective factor through psychotherapy and counseling intervention, which people often use without prescription to prevent or alleviate symptoms at onset. I would ask the authors if this was evaluated during the experiment or in the inclusion criteria; please add a short paragraph on this issue; if data are unavailable, add them to the limitations.
Response 1: Thank you for your comments. Factors such as psychotherapy and counselling intervention do affect the results of the experiment. In the manuscript, we restricted people who had taken psychotropic drugs or received antidepressant treatment in the past two months through exclusion criteria (page 3, lines 132-133). Since all subjects included in this study have never experienced treatment or intervention before, there are limitations in this study that have not been evaluated. As you suggested, we have added this limitation, as shown on page 21, lines 594-597 of the revised manuscript.
Reviewer 2 Report
The authors do a great job with this manuscript. I'd suggest a few edits
Introduction:
Overall the introduction is well written however, I'd recommend that the authors turn the last paragraph into not only proposing the hypothesis but presenting the final part of that paragraph as the objective of the study.
Methodology:
I would move the inclusion/exclusion criteria above where you state that the participants were split in HD and LD groups.
How many participants did you recruit? How many met your inclusion/exclusion criteria?
Please put 2.2 in past tense
How did you familiarize participants with this task? Was this done on a different day?
Please provide more information
When did participants complete the SDS scale?
Results:
Overall good job with presenting the results. Did you correct for multiple analyses?
Discussion
Based on the results the discussion is well written however, I need to gain a better understanding of the methodology.
Author Response
Dear reviewer,
Thank you for your valuable suggestions. We have revised the article in response to your suggestions. The following is the response to each proposal and the location of the changes:
Comments 1: Introduction: Overall the introduction is well written however, I'd recommend that the authors turn the last paragraph into not only proposing the hypothesis but presenting the final part of that paragraph as the objective of the study.
Response 1: Thank you for your suggestion. We have revised the language of this part of the manuscript to emphasize the last part of Introduction as the goal of this study. See page 3, lines 112-117.
Comments 2: Methodology: I would move the inclusion/exclusion criteria above where you state that the participants were split in HD and LD groups.
Response 2: Thank you for your suggestion. We have moved the inclusion/exclusion criteria after grouping participants. See page 3, lines 126-138.
Comments 3: How many participants did you recruit? How many met your inclusion/exclusion criteria?
Response 3: A total of 100 participants were recruited in this study, 62 of whom met the inclusion and exclusion criteria (30 in the high-risk depression group and 32 in the low-risk depression group). However, during the formal experiment, 4 participants were excluded due to incomplete data, so the final effective sample size of this study was 58. Thank you for your suggestion. We have completed the full subject recruitment process and changed the subject screening flow chart. See page 3, lines 120-123 and page 4, lines 146-147.
Comments 4: Please put 2.2 in past tense.
Response 4: Thank you for your suggestion. We have modified tense 2.2, see pages 4-7, lines 148-222.
Comments 5: How did you familiarize participants with this task? Was this done on a different day? Please provide more information.
Response 5: The tasks in this study are all free viewing tasks, and the researchers will inform the participants in detail of the process of completing the task before the subjects formally complete the task evaluation: "All you have to do is sit in a chair in front of the monitor and feel free to look at emotional photos that automatically appear on the monitor. The gray cross that flashes during each photo switch is eye movement calibration. The next photo will appear only when your eyes fall on the gray cross. Please feel free to look at all emotional photos that appear until the end of the paradigm." The test will be conducted immediately after the participants understand the task. In response to your suggestion, we have added more detailed information on the operation of the participants in the manuscript see page 4, lines 164 170.
Comments 6: When did participants complete the SDS scale?
Response 6: In our study, all participants completed the SDS scale prior to the "eye movement emotional perception evaluation paradigm". And the above evaluations were carried out on the same day to ensure the stability of the participants' anxiety state, to ensure the rigor of the test and to better clarify the effectiveness of this screening technique. Thank you for asking. In the manuscript, we have supplemented the time details of the participants' assessment of the SDS scale, see page 4, lines 173-177.
Comments 7: Results: Overall good job with presenting the results. Did you correct for multiple analyses?
Response 7: Your question is very professional. Due to our limited research conditions, this study only analyzed differences between the two groups of people in the high-risk and low-risk depression groups, and did not correct multiple analyses. We will look at the limitations, and our research has not been validated on the basis of multiple analyses. Thank you again for your professional advice, and we will further improve the data analysis results in future research.
Comments 8: Discussion:Based on the results the discussion is well written however, I need to gain a better understanding of the methodology.
Response 8: Thank you for your suggestion. We have reviewed and revised the content of the Discussion, see pages 19-20, lines 502-586.
Reviewer 3 Report
This is a well-structured article. The main question addressed by this research is the use of a new screening method, based on eye tracking technology, for risk of depression in young people.
The introduction gives the background of this study as it briefly describes depression and relevant diagnostic and screening tools.
“Materials and Methods” section is descriptive enough. It refers to the participants, the experimental design, the definition and quantitative analysis of digital biomarkers and the statistical analyses implemented during this study.
The results are quite interesting and, to my opinion, well presented and depicted in related figures.
The discussion is well written, summarizing and discussing the main findings of the study and trying to associate it with recent relative literature data. The fact that the authors have also included a paragraph about the main limitations of the study, to my opinion, adds to the scientific value of this paper.
Regarding “conclusions”, a section proposing some specific targets for future studies could be added.
References, although relatively few, are relative to the subject.
English language and style are generally fine but there are some minor issues that need to be addressed before publication (for instance, some long sentences could be separated in shorter ones, in order to make the text more comprehensible).
Author Response
Dear reviewer,
Thank you for your valuable suggestions. We have revised the article in response to your suggestions. The following is the response to each proposal and the location of the changes:
Comments 1: Regarding “conclusions”, a section proposing some specific targets for future studies could be added.
Response 1: Thank you for your suggestion. We have added specific objectives for our future research in the Conclusions section, see page 20, lines 602-606.
Comments 2: English language and style are generally fine but there are some minor issues that need to be addressed before publication (for instance, some long sentences could be separated in shorter ones, in order to make the text more comprehensible).
Response 2: Thanks for your suggestion. We reexamined the grammar and expression of the full text, and modified several long sentences in the manuscript to make the text easier to understand. See the page 1 lines 40-44, page 2 lines 55-60, page 3 lines 112-117, page 19 lines 502-505, page 19 lines 530-533, page 20 lines 571-574, page 20 lines 593-596.
Round 2
Reviewer 2 Report
I appreciate the authors addressing my concerns. My only suggestion would be to write that you did not correct for multiple analyses in your limitations section so there might be risk of a false discovery. On the other hand you could use a Benjamini Hochberg FDR with a 0.8 FDR rate (I ran a basic one and I don't think it would change your results) and still keep the results while correcting for multiple analyses.
Author Response
Dear reviewer,
Thank you for your valuable suggestions. We have revised the article in response to your suggestions. The following is the response to each proposal and the location of the changes:
Comments: I appreciate the authors addressing my concerns. My only suggestion would be to write that you did not correct for multiple analyses in your limitations section so there might be risk of a false discovery. On the other hand you could use a Benjamini Hochberg FDR with a 0.8 FDR rate (I ran a basic one and I don't think it would change your results) and still keep the results while correcting for multiple analyses.
Reply: Thank you very much for your suggestion. In order to reduce the risk of errors caused by non-correction, we have carried out Benjamini Hochberg to control FDR according to your suggestion. This is complemented by the Statistical Analysis section of Materials and Methods, see lines 341-342 on page 12 of the revised manuscript. And the FDR value after the correction of the P value is added to the Results table, see line 383 on page 13, line 396 on page 14, line 419 on page 14, line 438 on page 15.